# Mentorship in health research institutions in Africa: A systematic review of approaches, benefits, successes, gaps and challenges

**Maurine Ng'oda**[1]*, **Peter Muriuki Gatheru**[1,2], **Oyetunde Oyeyemi**[3], **Phylis Busienei**[1], **Caroline H. Karugu**[1], **Sharon Mugo**[1], **Lilian Okoth**[1], **Margaret Nampijja**[1], **Sylvia Kiwuwa-Muyingo**[1], **Yohannes Dibaba Wado**[1], **Patricia Kitsao-Wekulo**[1], **Gershim Asiki**[1], **Evelyn Gitau**[1]

**1** African Population and Health Research Center, Nairobi, Kenya, **2** School of Public Health, University of Ghana, Accra, Ghana, **3** Department of Biosciences and Biotechnology, University of Medical Sciences, Ondo City, Ondo Estate, Nigeria

* mngoda@aphrc.org, maurinekn2012@gmail.com

**Data Availability Statement:** Data underlying the findings for this review has been provided as part

## Abstract

### Background

In Africa, where the burden of diseases is disproportionately high, significant challenges arise from a shortage of skilled researchers, lack of research funding, and limited mentorship opportunities. The continent faces a substantial gap in research output largely attributed to the dearth of mentorship opportunities for early career researchers.

### Objective

To explore existing mentorship approaches, identify challenges, gaps, successes, and benefits, and provide insights for strengthening mentorship programs in African health research institutions.

### Methods

We registered the review protocol on the International Prospective Register of Systematic Reviews [CRD42021285018] and searched six electronic databases–EMBASE, AJOL, Web of Science, PubMed, DOAJ, and JSTOR from inception to 10 November 2023, for studies published in English reporting on approaches of mentorship in health research in African countries. We also searched grey literature repositories, institutional websites, and reference lists of included studies for additional literature. Two independent reviewers conducted screening of titles and abstracts of identified studies, full-text screening, assessment of methodological quality, and data extraction. We assessed study quality against the Mixed Methods Appraisal Tool (MMAT). We resolved any disagreements through discussion and consensus. We employed a narrative approach to synthesize the findings.

### Results

We retrieved 1799 articles and after screening, included 21 studies in the review. The reviewers identified 20 mentorship programs for health researchers (N = 1198) in 12 African

of the submitted article in the supplementary information.

**Funding:** This work was supported by the African Research Excellent Fund (AREF to EG). The funder had no role in study design, data collection and analysis, decision to publish, or preparation of the manuscript.

**Competing interests:** The authors have declared that no competing interest exist.

countries mostly focusing on early-career researchers and junior faculty members. A few included mid-career and senior researchers.

We categorized the programs under three key mentoring approaches: international collaborative programs, regional and in-country collaborations, and specialized capacity-building initiatives. Our review highlighted the following successes and benefits of health research mentorship programs: the establishment of collaborations and partnerships, development of research programs and capacities, improvement of individual skills and confidence, increased publications, and successful grant applications. The gaps identified were limited funding, lack of a mentorship culture, negative attitudes towards research careers, and lack of prioritization of research mentorship.

## Conclusion

Our review highlights a diverse landscape of health research mentorship aspects predominantly targeting early career researchers and heavily driven by the North. There is a need for locally driven mentorship initiatives in Africa to strengthen mentorship to advance health research in the region.

## Trial registration

PROSPERO registration number: CRD42021285018.

## Introduction

There exists a significant gap in research output in sub-Saharan Africa where the burden of disease is disproportionately high [1]. The current state of health science research, funding, and research capacity in the continent falls short of addressing the existing and unmet health research needs [2]. Some of the contributing factors to this challenge are the scarcity of well-trained and skilled researchers and the lack of opportunities for hands-on research experience with research specialists, leading to inadequate supervision and limited mentorship opportunities for early career researchers [3].

Mentorship is defined in simple terms as a relationship where someone experienced, in this context researcher, guides and supports another person to help them learn and grow professionally [4]. There are two common approaches to mentorship. The first approach is the traditional one-on-one mentoring model [5]. In this paradigm, an experienced researcher, often with a distinguished record of accomplishment, provides guidance and support to a less experienced mentee. This close, personalized interaction facilitates in-depth discussions, transfer of skills, and the cultivation of a strong mentor-mentee relationship [6]. Through this approach, the mentor can offer valuable insights, share experiences, and assist the mentee in navigating the complexities of the research landscape. The one-on-one model is particularly effective for tailoring mentorship to the unique needs and goals of the mentee, fostering a deep sense of individualized support and professional development [5].

The second common approach to mentorship in research involves group or team-based mentoring [7]. In this collaborative model, a mentor oversees a cohort of mentees who work together on related research projects or within a shared research theme. This approach promotes a sense of community and encourages peer learning among mentees.

Group mentoring can be especially beneficial in fostering interdisciplinary collaborations, providing diverse perspectives, and creating a supportive network for mentees [6]. It also allows the mentor's expertise to be leveraged across multiple individuals simultaneously [8]. The group dynamic enhances social learning, as mentees not only benefit from the mentor's guidance but also from the collective knowledge and experiences of their peers. Group mentorship is adaptable to various research settings and can effectively address the evolving needs of mentees in collaborative research environments [7].

Recognizing mentorship as a vital strategy for personal and professional growth [9,10], there is a growing awareness of its importance in enhancing the capacities of individuals, including researchers [11]. However, mentorship practices are not widespread in low- and middle-income countries (LMICs) [12], and available evidence on existing approaches demonstrates mixed results regarding effectiveness, often failing to account for the unique challenges and structures present in the research systems. The dearth of a well-established culture of mentorship, the absence of formal policies, and the inadequacy of structured tools for assessing mentorship further compound the challenges faced in fostering effective mentorship programs in LMICs [13].

The systematic review aimed to inform the development of optimized mentorship programs that address the specific needs and challenges faced by health researchers in Africa. We sought to synthesize evidence on various mentorship approaches prevalent in the region to provide a more comprehensive understanding of mentorship in health research institutions in Africa. We explored the nature and effectiveness of mentorship initiatives in African research institutions to identify both successes and challenges encountered in implementing these programs, pinpoint existing gaps in mentorship practices, and provide valuable insights.

The mentorship was considered effective if it resulted in early and mid-level career researchers taking up leadership roles and positions in research and academia, fostering a collaborative research environment, contributing to research outputs such as publications, and strengthening the skills of early and mid-level career researchers.

## Methods

### Protocol registration

We followed the Preferred Reporting Items for Systematic Reviews and Meta-Analyses (PRISMA) guidelines (2020) as shown in supplementary document (S1 Checklist). We registered the protocol for this review with the PROSPERO under the registration number CRD42021285018.

### Information sources and search

We identified relevant studies by searching various databases such as EMBASE, AJOL, Web of Science, DOAJ, PubMed, and JSTOR from their inception up to 10[th] November 2023. Additionally, we conducted searches on open grey literature repositories and specific institutional websites to identify any other relevant studies. We also conducted a manual search of reference lists of identified studies for any additional findings. A list of relevant search terms and keywords was prepared. The search terms were used in the following combinations: ("Practices") AND ("Success" OR "Benefits" OR "Advantages") AND ("Gaps" OR "Challenges") AND ("mentor" OR "mentorship" OR "mentoring" OR "mentoring relationship" OR "onsite training" OR "vertical mentorship" OR "on-the-job training" OR "OJT" OR "capacity building" OR "capacity strengthening" OR "mentee" OR "mentoring program" OR "mentoring models" OR "career coaching" OR "career counselling" OR "career support" OR "mentorship advice") AND ("research institutions" OR "research program" OR "researchers" OR "research

organizations") AND ("Africa" OR "African" OR "sub-Saharan Africa" OR "Africa South of the Sahara" OR "East Africa" OR "West Africa" OR "Southern Africa" OR "Central Africa" OR "Northern Africa"). To ensure that we retrieved articles from each country, we substituted the names of the study setting with the country-specific names. For example, to retrieve articles from East Africa, we replaced "East Africa" with Uganda, Kenya, Tanzania, Burundi, and Rwanda. The detailed search strategy is described in the supplementary documents (S1 Table).

## Study selection and eligibility criteria

Teams of two reviewers from MN, PMG, OO, SM, CHK, SM, LO, MNa, SKM, YDW, and PKW independently screened titles, abstracts, and full texts of the selected studies. Any disagreements between the two reviewers were addressed through discussion and consensus, or by consulting a senior reviewer (GA). The scope of our search was limited to studies published in the English language. We used the Population, Intervention, Comparison, Outcomes, and Study (PICOS) design as a framework to formulate eligibility criteria. The PICOS elements comprised; i) participants–researchers at any career level, serving as mentors or mentees; ii) interventions–diverse mentoring programs of varied types, durations, and regularities; iii) comparisons–all mentorship programs were considered, regardless of the presence of a comparison group; iv) outcomes–studies reporting on mentorship approaches, benefits, successes, gaps, and challenges were included in the review; v) settings–African academic and/or research institutions. Articles focusing on non-human health research were not eligible. We also excluded systematic reviews, conference abstracts, commentaries, and opinion pieces.

## Data collection process

Four reviewers (MN, PMG, OO, PB) independently extracted data from the selected studies using a Microsoft Excel extraction form. Key variables extracted were study author and date, country, study design, characteristics of the study population, sample size, intervention, mode of delivery, and outcome measures, including challenges, gaps, benefits, and successes. Discrepancies during the extraction process were resolved through discussion and consensus building.

## Assessment of methodological quality

Two reviewers (MN and PM) independently evaluated the quality of the included studies using the Mixed Methods Appraisal Tool (MMAT) [14], which enables the concurrent assessment of various empirical study types. The MMAT encompasses two general screening questions applicable to all study types and specific sets of five questions for each of the five study types: qualitative, quantitative randomized controlled trials, quantitative non-randomized, quantitative descriptive, and mixed methods design. Both reviewers utilized the MMAT criteria to assess key methodological components, including sampling, data collection, response bias, outcome measurements, and data analysis/reporting, providing a comprehensive evaluation of each study's overall quality. Disagreements were resolved through discussion and consensus. Ratings were assigned based on the proportion of fulfilled quality criteria, with studies classified as low risk ($\geq$75%), moderate risk (25–75%), or high risk (<25%). The included articles were categorised as qualitative, quantitative (observational), and mixed methods studies.

## Outcome measures

The primary outcome of this review was the mentorship approach including the mode and period of delivery. Secondary outcomes included successes, benefits, gaps, and challenges associated with the mentorship interventions.

## Synthesis of evidence

We employed a narrative synthesis approach to interpret findings from the included studies. For this reason, publication bias and heterogeneity in study designs, interventions, and outcomes were not considered. A comprehensive exploration of the outcomes of interest within the literature was achieved through a narrative synthesis. The synthesis involved summarizing the characteristics of included studies, such as study design, population, interventions, and key outcomes. We then categorized findings based on themes, similarities, and differences, providing a nuanced understanding of the evidence. The narrative synthesis was guided by the PRISMA guidelines.

# Results

## Search output

The initial search yielded 1623 articles from six databases. We identified an additional 176 articles through searches on open grey and specific institutional websites, resulting in 1799 retrieved articles. After removing 423 duplicates, we screened 1376 titles and abstracts, leading to the preliminary selection of 303 articles for full-text review. Ultimately, 21 studies that met the inclusion criteria were included in the review as shown in Fig 1 in the supplementary documents.

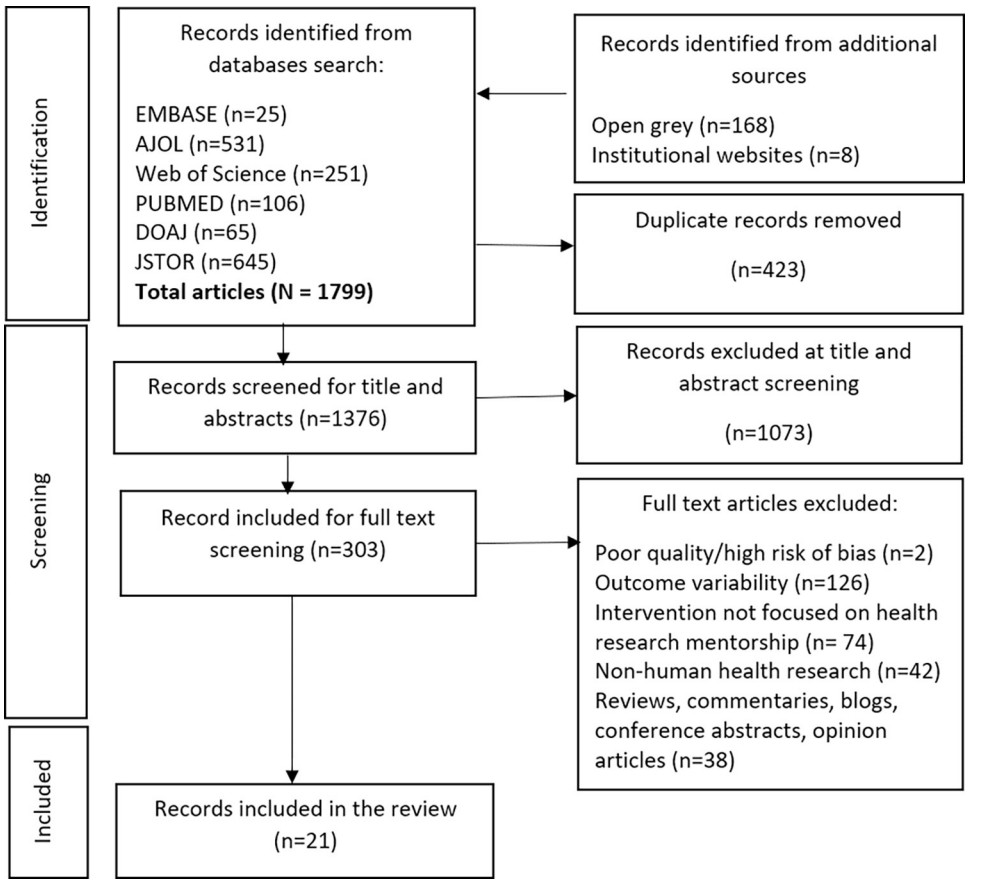

**Fig 1. Flow diagram for study identification.**

### Study characteristics

All the 21 studies included in the review were observational studies published between 2013 and 2023. The studies were from 12 different African countries including Kenya, Tanzania, Uganda, Ethiopia, Nigeria, South Africa, Malawi, Ghana, Liberia, Zimbabwe, Rwanda, and Lesotho. The primary studies used diverse methods which included qualitative designs (10 studies; 48%) [15–24], quantitative designs (9 studies; 43%) [25–33], and mixed methods (2 studies; (9%) [32,33].

### Characteristics of participants

The review included a diverse cohort of early-career, mid-career, and senior researchers (N = 1198) from various institutions. The participants were recruited from universities [13,14,18,19,21,24,26,27,29,30,32], health research institutions [15–17,20,22,23,25,28], public health teaching institutions [33], and hospitals [35]. Notably, their work focus spanned a range of fields, including HIV/AIDS rsearch, mental health, sanitation and hygiene, family health, biomedical sciences, biostatistics, health system and policy, and public health. The inclusion of mid- or senior-level faculty researchers, doctoral fellows, statisticians, and undergraduate students contributed to a well-rounded participant pool. This diversity not only enriches the study's findings but also underscores the broad relevance and applicability of research mentorship across multiple disciplines within the health sciences. The detailed characteristics of the individual studies included in the review can be found in supplementary information (S2 Table).

### Methodological quality of individual studies

Ten qualitative studies, nine quantitative studies, and two mixed-methods studies were assessed for methodological quality. We rated eight studies as low risk and 13 studies as moderate risk. Detailed information on the risk rating for each study can be found in supplementary information (S3 Table).

### Approaches of research mentorship

We identified three key approaches from the included studies that have been used for mentoring health researchers in Africa. Broadly, we have categorized these into international collaborative programs [14–16,19,24,33], regional and in-country collaborations [17,18,20,23,31], and specialized capacity-building programs [21,22,25–28,30,32,34].

**International collaborative programs.** International collaboration emerged as a central theme in the findings as one of the approaches used in health research mentorship, demonstrating a concerted effort to foster exchanges through cross-cultural training programs global symposia and workshops, resources sharing through north-south and south-south collaborations, and building global community of researchers through multinational research hubs, global networks, and infrastructure development. These collaborative initiatives aimed to transcend geographical boundaries, leverage diverse expertise, and collectively address health research challenges to achieve sustainable and impactful outcomes.

The ARCADE project [20] and the 5-year multinational collaboration across five African countries, the USA, and the UK highlight both north-south and south-south collaborations [34]. These initiatives brought together researchers from different continents, acknowledging the importance of shared expertise and resources. The collaborative capacity strengthening initiative at the University of Western Cape (UWC) in South Africa [16], involving international symposia and workshops provided platforms for researchers to come together, share insights,

and engage in collaborative learning. The exchange of ideas fosters a global perspective on research challenges, methodologies, and solutions. The AIDS International Training and Research Program (AITRP) [15] and the collaborative capacity strengthening initiative involving the UWC in South Africa [16] exemplify cross-cultural training programs.

**Regional and in-country collaborations.** In-country and regional collaborations are also prominently highlighted as an avenue to mentorship, reflecting the recognition of the importance of strengthening research capacity at the local and regional levels. This theme involves partnerships and initiatives that focus on collaboration within a specific country or region. For instance, the Nigeria Implementation Science Alliance (NISA) [19], an initiative that involves collaboration among partners within Nigeria and the sub-Saharan African region focuses on local research capacity strengthening. The program aims to facilitate collaboration, enhance implementation research, and identify culturally appropriate strategies to improve public health through research.

The Transforming Health Professions Education in Tanzania (THET) project [32] included a component where young peers received mentorship from senior researchers through mentored research awards and research training. These peers, in turn, provided reciprocal peer-to-peer mentorship to undergraduate students.

This approach emphasizes the importance of building mentorship networks within the country, creating a sustainable model for capacity strengthening. In a separate example, a series of two-day intensive regional mentorship workshops were conducted over four years to train mid- and senior-level investigators engaged in public health, clinical, and basic science research across multiple academic institutions in LMICs [21]. These workshops focused on developing mentorship skills locally and regionally, recognizing the value of nurturing research talent within specific geographic contexts. The African Mental Health Research Initiative (AMARI) [23] recruits and trains research fellows at Master's, PhD, and post-doc levels within the African region. The initiative aims to equip these fellows with research, teaching, and leadership skills to build a viable and sustainable research network.

**Specialized capacity-building programs.** This approach recognizes the importance of tailoring mentorship initiatives to the unique needs and challenges faced by researchers in Africa and involves targeted initiatives designed to enhance specific skills and competencies related to health research. For example, the Consortium for Advanced Research Training in Africa (CARTA) program [26], delivered through PhD training fellowships is a specialized training that focuses on creating a network of locally trained but globally recognized African scholars. CARTA recognizes the importance of advancing research capacity at the doctoral level locally, contributing to the development of a cadre of highly skilled researchers.

The Sanitation and Hygiene Applied Research for Equity (SHARE) program [28] incorporates specialized mentoring integrated into research, administration, financial management, and communication activities. This approach ensures that participants receive guidance and support in areas directly relevant to their research projects.

The Sexual Violence Research Initiative [18] provided intensive mentoring and technical advice specifically for the development or adaptation and conduct of preliminary proof of concept testing of violence against women and violence against children primary prevention interventions. This specialized training addressed the unique challenges associated with research on sensitive topics and provided targeted support for researchers in the field of sexual violence prevention. The Medical Education Partnership Initiative–Medical Education for Equitable Services for All Ugandans (MEPI-MESAU) program [29] goes beyond the traditional mentorship by providing infrastructure support including administrative support, paid tuition fees, tools, and skills training–on study design, biostatistics, manuscript and grant writing, to early career researchers.

Lastly, initiatives like AFFIRM, LATIN-MH, PAM-D, RedeAmericas, and SHARE, funded by the National Institute of Mental Health (NIMH), specifically targeted mental health research [24]. These hubs aimed to improve the research core for evidence-based mental health interventions, enhance research skills in global mental health, and provide capacity-building opportunities for early career investigators in LMICs. In Rwanda, the 6-week deliverable-driven survey analysis training program [27] aimed at strengthening the skills of local research leaders and statisticians. This hands-on training focused on a specific aspect of research (survey analysis) and was designed to achieve tangible outcomes within a defined period. S4 Table summarizes the approaches to mentorship in health research identified in the 21 studies included in this review.

## Successes

In this section, we highlight the diverse successes derived from the health research mentorship programs implemented in 12 different African countries. We consolidate these successes into five crucial themes outlined herewith: establishment of partnerships and collaborations; development of research programs; individual capacity strengthening; development of research publications; and successful grant applications and awards.

**Establishment of partnerships and collaborations.** Six studies mentioned the establishment of partnerships and collaborations as one of the successes of mentorship programs in health research institutions in Africa [15,19,24,25,28,29]. The successes related to this aspect included the establishment of mutually beneficial collaborations between investigators in different countries that were developed during training, which built a supportive research environment. There were also shared and mutually beneficial resources within international research collaborations, which supported early career investigators and served as a conduit to transfer health research training opportunities to researchers in African institutions [15]. Through mentorship programs, various organizations and government agencies were able to make definite commitments toward more investment in implementation research. For example, in Nigeria, the National Agency for the Control of AIDS (NACA) in collaboration with the United Nations Children's Fund (UNICEF) and the Population Council was able to launch the "Adolescent and Young People Challenge" pilot project. This project sought to fund innovative ideas led by youth to provide comprehensive HIV education to at least 200,000 Nigerians [19]. The mentorship programs also led to the establishment of sustainable partnerships between researchers in sub-Saharan African countries and other LMICs, as well as with institutions in the north.

These partnerships facilitated collaborative cutting-edge research in global mental health and provided a management strategy that builds partnerships between local and international partners for efficient coordination and timely achievement of set goals [25].

**Development of research programs.** Thirteen studies reported the development of research programs as a key success of the respective mentorship programs. The different aspects achieved under this theme, as reported by the highlighted studies, were that early career investigators learned how to navigate the complex international research environment to build local research capacity [14,16] with trainees experiencing moderate increases in research confidence that were statistically significant, and an observed positive research culture being created [35]. In a study conducted in Uganda, the mentored PhD students were able to supervise and mentor 65 Master's students, thus building local research capacity [28]. Participation in workshops provided knowledge of valuable concepts and a structure for the development and strengthening of formal mentoring programs across LMIC institutions, leading to the growth of institutional support, the establishment of several new institutional mentorship

training programs, the initiation of peer mentorship networks, and regular mentor-mentee meetings. A qualitative study conducted in Kenya, Peru, India, and South Africa reported that the mentorship training model expanded as a national mandate for research training, nested within a required training program [21]. Hubs that evolved into centers of research excellence with a crop of dedicated researchers were also established [25].

**Individual capacity strengthening.** Individual capacity strengthening was reported to increase as participants engaged in various training programs, workshops, and research activities.

For instance, in Zimbabwe, faculty members attended at least one of 15 faculty development workshops. Forty-one faculty members underwent a one-year advanced faculty development training in medical education and leadership, 33 mentored research scholars were trained under the novel NECTAR, and 52 and 12 in cardiovascular and mental health programs, respectively [34]. In Rwanda, three-quarters of the participants mentored others in survey data analysis or conducted an additional survey analysis in the year following the training. Similarly, 36% of participants completed an additional DHS analysis, 71% completed an additional survey analysis, and 79% provided mentorship to others about survey data analysis [27].

In addition, individual capacity strengthening was achieved as mentors enrolled in other courses or training. In Tanzania, most young peers had taken at least three research training short courses, and six had enrolled in PhD programs. The number of fellows increased from 12 to 24, and mentored graduates increased from 41 to 67 in the second cohort. Eight senior fellows enrolled in PhD programs, and 10 of 12 had registered for a PhD fellowship [31–33]. In Malawi, Uganda, and South Africa, the ARCADE project was successful in developing and delivering courses that reached over 920 postgraduate students [20]. In Ethiopia, Ghana, Malawi, South Africa, Uganda, Zimbabwe, Kenya, Liberia, Nigeria, and Sri Lanka, the mentorship programs have achieved more success, including participants completing their courses for second master's degrees with a special focus on specific health aspects, winning awards to support the further development of their research careers, and the appointment of one participant as a professor and another young researcher at a Health Institute [24].

**Development of research publications.** The mentorship programs led to the development of new publications in various fields, as reported by nine studies. For instance, during the first two years of the program in Tanzania, various research articles from the mentored programs were published, with other manuscripts in the final stages of preparation. Each mentee had at least one or more manuscripts published or accepted for publication, and young peers shared authorship in at least one of the published articles [31–33]. Various publications were also done in other different mentorship programs across various countries, with authorship being from multi-institutional teams and submitted to international peer-reviewed scientific journals [23,24,26–28]. In South Africa, 70 interns contributed to 51 peer-reviewed articles [22], while in Uganda, 80 publications not related to PhD thesis work were co-authored by PhD students [29].

**Successful grant applications and awards.** Five studies reported funding applications with some grants being awarded as a major success derived from the mentorship programs. A study done in Ghana, Kenya, Liberia, Nigeria, and South Africa reported 21 grant applications being made successfully over the mentorship period [25]. Similarly, in Tanzania, young peers began to broaden their research careers by involvement in other ongoing research projects and grant applications [33], with a majority (n = 7/12) receiving research grants for their research program [32] and six small- to medium-sized research grants won [31]. Similarly, the mentorship programs led to the formation of a peer network of researchers that was deemed a pivot of success.

For instance, through the SHARE program, nine networks were created during phase II of the project, out of which six of the PhD students have pursued research that has led to independent grant funding, as well as collaborative grants on which they are listed as a co-investigators [29]. The mentorship programs also led to awards. For instance, two travel fellowship grants for early career researchers to attend the 2016 and 2017 World Psychiatric Association International Congress were won. Grants to attend conferences to share findings for completed dissertation projects were won, enabling participants to interact with other external partners and build sustainable collaborations [24].

## Benefits

This review reveals benefits that extend beyond the individual participants. We summarize these benefits under three pivotal themes: capacity building and skill development; networking and collaboration; and career advancement and marketability.

**Capacity building and skill development.** The mentorship programs led to significant skill development and capacity building among participants [26,32]. This was evident through diverse training in research methodologies, epidemiology, biostatistics, grant writing, and other crucial aspects [33]. The acquisition of these skills not only enhanced the participants' ability to conduct high-quality research [28,33] but also made them valuable contributors to national and international projects [17]. The establishment of training centers and departments further institutionalised these skills, fostering a culture of continuous learning and research excellence [26,28].

**Networking and collaboration.** The initiatives for research mentorship played a crucial role in fostering strong collaborations among institutions and researchers [20]. These collaborations were instrumental in the success of various projects and contributed to the publication of research papers [24,28]. The projects served as a platform for early career and mid-level researchers to take leadership roles in published papers [25], demonstrating the effectiveness of mentorship in fostering a collaborative research environment. In addition, the North-South and regional collaboration programs exposed participants to international perspectives thus encouraging the integration of local and global knowledge [16,33].

**Career advancement and marketability.** There was a positive impact of the mentorship programs on the career trajectories of participants [28,32,33]. Interns who engaged in significant research projects became more marketable as research practitioners [22]. The experience gained and the demonstrated completeness of their work opened doors to attractive positions in academia [32,33], government, and other sectors [22]. Additionally, the model of mentorship proved effective in strengthening skills among full-time working professionals [27], allowing them to enhance their capabilities without disrupting ongoing work commitments. This contributed to the overall growth of faculty [34], increased student enrolment [33], and the establishment of new research support centers [34].

## Challenges and gaps

The reviewed studies identified limited funding and the absence of a robust mentorship culture as significant barriers to research advancement.

Negative perceptions of research as a career path and the lack of emphasis on mentorship in research further exacerbated the reported obstacles. The challenges were compounded by the COVID-19 pandemic, which disrupted research operations and constrained available resources. These factors are discussed in detail in the following section.

**Limited funding.** Six studies [15,16,18,23,25,32] reported on the challenges of funding for health research mentorship programs. Limited funding encompassed the failure of health

researchers to access funding to support research, the inability of early career researchers to access independent research funding without external collaborators, and the inability to secure long-term funding for meaningful capacity strengthening. Limited funding was also reported to include a lack of support for degree programs, post-doctoral training, and research [14,25]. A mismatch between the availability of short-term funds for specific research initiatives and requirements for longer-term investment in capacity building was reported as a gap [16]. Lastly, in instances where funding is available, the funders often drive the focus of mentorship programs, and the lack of southern ownership was identified as a gap [16].

**Lack of a healthy mentorship culture.** In seven studies, [15,20–23,25,27], the lack of understanding of the concept of mentorship leading to a lack of institutional mentoring culture was highlighted as a challenge. Sustaining mentorship and institutional support for mentorship, and failure by institutions to acknowledge or 'give credit' for mentoring activities in the merit or promotional processes are notable challenges in health research mentorship.

Of particular concern in many of the mentorship programs was a general lack of time management strategies to balance mentoring with other competing activities including academic pursuits, teaching duties, and burdensome administrative roles. Related to the lack of a mentorship culture were limited mentoring skills, and a lack of motivation, or zeal on the part of both mentors and mentees. Mentors reported that getting mentees to understand their roles and commit to achieving the set goals was a burdensome challenge [16]. Mentorship was also reported to place a heavy demand on senior researchers' time [21], which is already committed to urgent needs such as obtaining donor funds, reporting to donors, managing projects, networking, and publishing–all attached to a researcher's performance appraisal. This in turn led to increasing levels of stress among mentors and very little time left to focus on mentees who needed significant guidance and support [28].

For mentees, different sets of administrative regulations across institutions were reported to lead to complications and delays in starting or sustaining certain capacity-building activities. Mentees reported difficulties in balancing work burdens, as they were involved in research activities as well as the training and support for their institutions and their development. This lack of protected time for health research mentorship was also cited as a gap by one study [33]. Lack of infrastructure support that enables high-quality research including grants administration, mentorship, research leadership, research culture, and open communication between policymakers and researchers as well as difficulties in accessing a PhD supervisor were other challenges faced by mentees [20–23]. Several other gaps were also identified including the lack of recognition of mentorship as a key success factor for early career researchers [21], the absence of a formal mentorship structure [21,26], and the lack of clarity in expectations of a mentor-mentee relationship [21].

**Negative attitudes towards research as a career.** Research, as a career, was not a very attractive proposition in many Southern contexts according to some studies [16–19]. Researchers being drawn by incentives to 'consultancy not research' complicated this. Research was also viewed as inaccessible, especially to young people and outside academic settings. Lack of research interest was cited as a gap in three studies [14,15,22], with institutions such as universities or health departments prioritizing teaching rather than focusing on research careers [18,29].

**Lack of prioritisation of research mentorship.** The low priority given to research mentorship by funders and governments was recognized in studies conducted in South Africa, Rwanda, Ghana, Kenya, Liberia, and Nigeria [21,24], and weak collaborations between different stakeholders and countries involved in mentorship may have contributed to this [16]. The absence of a national research strategy [16] was also identified as a gap in research mentorship programs in various African countries.

**COVID-19 related factors.**   Challenges related to the emergence of COVID-19 were reported in two studies [31,32] and included halting physical meetings between mentors and mentees because of the global restriction of face-to-face meetings. Other COVID-19-related challenges included the suspension of research activities such as enrolment of participants, procuring of laboratory reagents, delays in data collection, hiking of prices, and delays in delivery of procured research materials [32]. Internet connectivity challenges leading to suboptimal quality of video conferences were also highlighted as challenges [33].

## Discussion

In this review, we identified 20 mentorship programs involving a diverse group of African health researchers across 12 countries in sub-Saharan Africa. Only two of these were initiated in Africa and funded from local sources. While most African researchers have benefited from North-South collaboration, there is an opportunity to develop local mentorship programs to reduce the overreliance on foreign-funded and foreign-driven programs. Foreign-initiated and driven programs can be beneficial to building local health research capacity; however, local programs are often more accessible and sustainable, given their understanding of the local, context, infrastructure, and resources [30]. Such programs can foster a stronger sense of community and collaboration, contributing to long-term impact. Locally led initiatives also empower African mentors to play leadership roles, reinforcing a sense of ownership and self-determination in shaping the future of their communities [26]. Overall, locally initiated mentorship programs are better positioned to address the nuanced localized needs of mentees, promoting a more inclusive and impactful approach to personal and professional development. This finding is similar to those reported in earlier studies [36] that most health research mentorship initiatives in LMICs were introduced and funded by high-income countries and were not institutionalized as yet. Nevertheless, even though few, Africa-led, Africa-centered, and Africa-specific initiatives such as the Alliance for Accelerating Excellence in Africa (AESA) and the Coalition for Research and Innovation (CARI) are platforms that can provide support for training African researchers and opportunities for collaboration [1].

Our review highlights mentorship benefits that extend beyond the individual level to institutional, country, regional, and international arenas. Capacity building and skills development, networking and collaboration, and career development and marketability were highlighted in the reviewed studies.

Not only were individuals participating in mentorship programs upskilled in various aspects such as research methodologies, epidemiology, biostatistics, and grant writing among other skills, but mentorship enabled individuals to contribute to national and international projects. Enhancing individual capacities enables local researchers and junior faculty to navigate the complex international research environment and transfer health research training to African institutions. Even though most mentorship initiatives are North-initiated and driven, the programs expose participants to international perspectives that contribute to the integration of local and international knowledge. Additionally, participants are also enabled to develop their research niches within academia, government, and the private sector.

We further establish that the main hindrance to the development of health research capacity including mentorship programs is limited local funding. Current funding for health research and research capacity development remains inadequate to address Africa's unmet health needs. This calls for African countries to develop clear and context-informed strategies and mechanisms to foster both private and public investment in health research capacity development. Furthermore, African countries can leverage international programs that can be institutionalized and tailored to respond to local needs for health research capacity development.

Consistent with our findings, limited local health research capacity development funding has also been previously highlighted by other researchers as a major challenge to capacity development [1,14,35].

Additionally, the lack of a healthy mentorship culture in most African health research institutions mostly arising from a lack of understanding of the concept and importance of mentorship in research capacity development was a substantial gap. Efforts are needed to ensure that mentorship is appreciated and given credit during merit and promotion activities.

This will ensure that mentorship is prioritized alongside other core research capacity activities such as teaching, administrative roles, applying for grants, managing projects, reporting to donors, and networking. Coupled with mentorship prioritization, capacity development in mentorship skills, and arousing interest to engage in mentorship would also contribute to mentorship being treated as an important part of health research capacity development.

Lastly, for mentorship to be viewed as a key success factor for early career researchers and junior faculty, institutions must address unfavorable administrative regulations, and lack of protected time along the mentorship continuum for both mentors and mentees. Additionally, deliberate efforts to establish formal mentorship structures, provide clarity in expectations of a mentor-mentee relationship, and prioritize mentorship on the part of funders, governments, and institutions will go a long way in institutionalizing health research mentorship in Africa.

## Conclusion

Our review revealed three main approaches to mentorship in Africa: international collaborative programs, regional and in-country collaborations, and specialized capacity-building programs. The successes of these programs were diverse and included the establishment of partnerships, the development of research programs, individual capacity strengthening, increased publication outputs, and successful grant applications and awards. These programs not only benefited individual participants but also contributed to broader capacity building, skill development, networking, collaboration, and career advancement at institutional, country, regional, and international levels.

However, several challenges and gaps were identified, such as limited funding, a lack of a healthy mentorship culture, negative attitudes toward research as a career, lack of prioritization of research mentorship, and challenges related to the COVID-19 pandemic. The review emphasizes the critical need for increased local funding for health research mentorship programs, the establishment of a robust mentorship culture, and addressing challenges related to administrative regulations, protected time, and mentorship skills. Furthermore, the findings underscore the importance of developing locally initiated mentorship programs to reduce reliance on foreign-funded initiatives. Researchers should make efforts to establish local and regional collaborative partnerships. While international collaborations are valuable, locally-led programs can be more accessible, sustainable, and tailored to address nuanced local needs, fostering a stronger sense of community and collaboration.

In addressing the identified challenges and building on the successes, African countries must develop clear and context-informed strategies for both public and private investment in health research capacity development. Additionally, efforts are needed to promote mentorship appreciation in merit and promotion activities, develop mentorship skills, and institutionalize mentorship structures. Only through these comprehensive efforts can health research mentorship be prioritized and effectively contribute to the sustainable development of research capacity in Africa.

## Limitations of the study

While this systematic review provides important insights into mentorship programs for health researchers in African institutions, it is crucial to recognize certain inherent limitations in the

study design and execution. The review's focus on studies published exclusively in English introduces a potential language bias, as pertinent research in other languages might have been overlooked, potentially impacting the thoroughness of the findings. Additionally, despite efforts to include diverse African regions, the search strategy may exhibit biases toward specific countries or regions, stemming from variations in research visibility and accessibility. This potential bias could constrain the generalizability of the findings across the entire African continent. To mitigate these limitations, multiple databases were consulted, and searches were conducted using a variety of relevant keywords and MeSH terms to retrieve as many articles as possible and to ensure a comprehensive coverage of mentorship programs across different African contexts. Additionally, we did not synthesize data regarding the most successful mentorship approaches in terms of the outputs so this topic should be explored in future reviews. Lastly, the heterogeneity of mentorship programs, characterized by variations in types, durations, and regularities, poses challenges in comparing and synthesizing outcomes. To address this challenge, we systematically categorized and classified mentorship programs based on predefined criteria, allowing for a structured synthesis of findings. Despite these limitations, the systematic approach, adherence to PRISMA guidelines, and comprehensive exploration of mentorship outcomes enhance the credibility of the findings derived from this review. Nevertheless, researchers and policymakers should approach the interpretation of the results with an awareness of these limitations and endeavour to conduct further research addressing identified gaps and challenges in mentorship programs for health researchers in Africa.

## Supporting information

**S1 Checklist. PRISMA 2020 checklist.**
(DOCX)

**S1 Table. Search stratey.**
(DOCX)

**S2 Table. Characteristics of studies included in the review.**
(DOCX)

**S3 Table. Risk of bias assessment.**
(DOCX)

**S4 Table. Mentorship approaches.**
(DOCX)

**S5 Table. List f abbreviations.**
(DOCX)

**S6 Table. Studies retrieved.**
(XLSX)

**S7 Table. Data extracted.**
(XLSX)

## Author Contributions

**Conceptualization:** Maurine Ng'oda, Peter Muriuki Gatheru, Oyetunde Oyeyemi, Phylis Busienei, Caroline H. Karugu, Sharon Mugo, Lilian Okoth, Margaret Nampijja, Sylvia Kiwuwa-Muyingo, Yohannes Dibaba Wado, Patricia Kitsao-Wekulo, Gershim Asiki.

**Data curation:** Maurine Ng'oda, Peter Muriuki Gatheru, Oyetunde Oyeyemi, Phylis Busienei, Lilian Okoth, Margaret Nampijja, Sylvia Kiwuwa-Muyingo, Yohannes Dibaba Wado, Patricia Kitsao-Wekulo, Gershim Asiki.

**Formal analysis:** Maurine Ng'oda, Peter Muriuki Gatheru, Phylis Busienei.

**Funding acquisition:** Lilian Okoth, Patricia Kitsao-Wekulo, Gershim Asiki, Evelyn Gitau.

**Investigation:** Maurine Ng'oda, Peter Muriuki Gatheru, Oyetunde Oyeyemi.

**Methodology:** Maurine Ng'oda, Peter Muriuki Gatheru, Oyetunde Oyeyemi, Patricia Kitsao-Wekulo, Gershim Asiki.

**Project administration:** Lilian Okoth, Patricia Kitsao-Wekulo, Gershim Asiki, Evelyn Gitau.

**Resources:** Lilian Okoth, Gershim Asiki.

**Supervision:** Maurine Ng'oda, Patricia Kitsao-Wekulo, Gershim Asiki.

**Validation:** Maurine Ng'oda, Gershim Asiki.

**Writing – original draft:** Maurine Ng'oda, Peter Muriuki Gatheru, Phylis Busienei.

**Writing – review & editing:** Maurine Ng'oda, Peter Muriuki Gatheru, Oyetunde Oyeyemi, Phylis Busienei, Caroline H. Karugu, Sharon Mugo, Margaret Nampijja, Sylvia Kiwuwa-Muyingo, Yohannes Dibaba Wado, Patricia Kitsao-Wekulo, Gershim Asiki.

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
