## [Decision Letter · Decision Letter 0]

2 Jul 2024

PGPH-D-24-01117

Mentorship in Health Research Institutions in Africa: A Systematic Review of Approaches, Benefits, Successes, Gaps and Challenges

Dear Dr. Ng'oda,

Thank you for submitting your manuscript to PLOS Global Public Health. After careful consideration, we feel that it has merit but does not fully meet PLOS Global Public Health’s publication criteria as it currently stands. Therefore, we invite you to submit a revised version of the manuscript that addresses the points raised during the review process.

We look forward to receiving your revised manuscript.

Kind regards,

Janet Seeley

Academic Editor

Journal Requirements:

2. Please provide separate figure files in .tif or .eps format only and remove any figures embedded in your manuscript file. Please also ensure all files are under our size limit of 10MB.

Additional Editor Comments (if provided):

Thank you for this interesting paper. The reviewer has some suggestions for improvement which I believe will strengthen your paper. I realise that re-running the review with individual African country names may seem to be a large task - but I would urge you to run a check to see that nothing has been missed. You can then update your methods to assure readers that this was done. If you find papers that were missed, they can be added in and you can explain in your methods that you ran a country search separately to ensure that no papers were missed.

Reviewers' comments:

Reviewer's Responses to Questions

**Comments to the Author**

1. Does this manuscript meet PLOS Global Public Health’s publication criteria? Is the manuscript technically sound, and do the data support the conclusions? The manuscript must describe methodologically and ethically rigorous research with conclusions that are appropriately drawn based on the data presented.

Reviewer #1: Yes

2. Has the statistical analysis been performed appropriately and rigorously?

Reviewer #1: N/A

3. Have the authors made all data underlying the findings in their manuscript fully available (please refer to the Data Availability Statement at the start of the manuscript PDF file)?

Reviewer #1: Yes

4. Is the manuscript presented in an intelligible fashion and written in standard English?

Reviewer #1: Yes

5. Review Comments to the Author

Reviewer #1: This is a very well written systematic review of the landscape of mentorship across African institutions. The authors have described their systematic approach to the study, which was registered on PROSPERO and follows the PRISMA guidelines and the PICO approach applied using MMAT for reviews of mixed methods studies. The review outlines the key benefits and challenges related to different mentorship models for capacity strengthening in health research in Africa. I only have a couple of minor comments on the paper:

1. The initial search term do not include lists of individual country names and I wonder if this has led to missing some initiatives from the study

2. The authors explicitly state that the review will consider effectiveness but do not provide a clear definition of what effectiveness means in this context. Inclusion of this would substantially enhance the review.

3. This is perhaps beyond the scope of this review but I would be interested to see how capacity strengthening impacted African researchers by gender, I wonder if the authors might comment on this at least?

4. I also wonder whether the authors might be able to comment in the conclusions on which approaches were considered most successful in terms of outputs based on mentorship approaches - this would substantially increase the importance of this paper

These are simply recommendations for minor revisions, to move towards greater interpretation of the findings, but the paper is well structured, clearly thought through and very well written overall.

6. PLOS authors have the option to publish the peer review history of their article (what does this mean?). If published, this will include your full peer review and any attached files.

**Do you want your identity to be public for this peer review?** For information about this choice, including consent withdrawal, please see our Privacy Policy.

Reviewer #1: No

---

## [Editor Report · Decision Letter 1]

27 Aug 2024

Mentorship in Health Research Institutions in Africa: A Systematic Review of Approaches, Benefits, Successes, Gaps and Challenges

PGPH-D-24-01117R1

Dear Dr Ng'oda,

We are pleased to inform you that your manuscript 'Mentorship in Health Research Institutions in Africa: A Systematic Review of Approaches, Benefits, Successes, Gaps and Challenges' has been provisionally accepted for publication in PLOS Global Public Health.

Best regards,

Janet Seeley

Academic Editor

Thank you for your attention to the comments - and the changes you have made. I do hope this paper is read widely - I look forward to seeing this published.